# Service user perceptions of smoking cessation in residential substance use treatment

**Zoe Swithenbank**[ID]*, **Rebecca Harrison**[ID], **Lorna Porcellato**

Public Health Institute, Liverpool John Moores University, Liverpool, United Kingdom

* z.a.swithenbank@2019.ljmu.ac.uk

## Abstract

### Introduction

Prevalence of tobacco smoking among adults in substance misuse treatment is much higher than the wider population, yet limited research is available, and residential treatment services have been overlooked as a potential setting for cessation interventions. Exploring the perceptions of service users about smoking cessation in residential rehabilitation is important to gain better understanding of this issue and identify ways to inform future intervention development.

### Methods

Ten semi-structured interviews were conducted in the Northwest of England in 2017 with adults (7 male, 3 female) who were currently or had previously been in residential treatment for substance misuse. Five participants were current smokers, three had never smoked, and two were former smokers. Participants were asked about their smoking behaviours, factors relating to smoking and smoking cessation and the relationship between smoking and substance use. All interviews were transcribed and data was analysed thematically.

### Results

Study findings highlighted a general consensus amongst participants that residential treatment services offered an ideal opportunity for cessation but there were concerns that doing so might jeopardise recovery. Smoking in substance use treatment services is still the norm and factors such as perceived social and psychological benefits, normative behaviours and lack of perceived risk or prioritisation pose challenges for implementing smoking cessation within this setting, although facilitators such as motivation to change and appropriateness of the setting were also identified.

### Conclusions

This study suggests that service users perceive residential treatment services as suitable environments to introduce smoking cessation. To address the needs of adults who smoke and are in recovery from substance use, further research and cooperation from treatment organisations is needed to integrate substance misuse and smoking cessation services.

**Data Availability Statement:** Data cannot be shared publicly due to the vulnerable nature of the participants. Data that are not directly identifying may also be inappropriate to share, as in this case a small number (10) of participants were involved.

Data are available from the Liverpool John Moores University Institutional Data Access / Ethics Committee (contact via D.Harriss@ljmu.ac.uk) for researchers who meet the criteria for access to confidential data.

**Funding:** The author(s) received no specific funding for this work.

**Competing interests:** The authors have declared that no competing interests exist.

More conclusive evidence on the effectiveness of tackling both issues at the same time is also required.

## Introduction

Tobacco use carries a significant mortality of an estimated eight million people per year, worldwide.

[1]. Smoking in adults in substance use treatment is highly prevalent, estimated at between 74% and 98% [2], putting them at a much greater risk of smoking related death or associated health conditions [3]. Evidence also suggests that smoking is one of the key causes of death in formerly alcohol or dependent adults [4]. Despite this, smoking amongst alcohol and drug dependent adults is largely ignored and untreated [5, 6].

The 2017 UK Government Drugs Strategy [7] recognised the significance of smoking in drugs and alcohol using populations and suggested that treatment providers should work with smoking cessation services to offer cessation and harm reduction. Reid et al. [8] observed improved outcomes when cessation was offered in community settings and Sees and Clark [9] reported high rates of patients interested in smoking cessation at the early stages of substance use treatment. Patients entering substance use treatment are ready to engage with treatment and have demonstrated a commitment to change, as evidenced by Prochaska and DiClemente's Transtheoretical model [10]. This can also offer a 'teachable moment' [11] where clients are at a greater likelihood of implementing positive behaviour change.

A common perception is that treatment for tobacco addiction can jeopardise the success of recovery from other substances and should be offered once one addiction is treated [12]. Joseph et al. [13] noted that treatment staff had been advising clients to avoid smoking cessation as this would jeopardise their recovery from the primary substance. Others however have challenged this, suggesting that smoking cessation does not impair outcomes for the presenting substance misuse problem [14–16]. Richter and Arnsten [17] suggest that outcomes for both substances may be improved whilst Weinberger et al. [18] argue that continued smoking or smoking initiation among non-smokers are associated with greater odds of a relapse to the use of the primary substance, and that incorporating smoking cessation into substance misuse treatment may improve long-term substance use outcomes.

Despite evidence to support the need for cessation in substance use treatment, little research is available on service user perspectives. Grigsby, et al. [19] argue that more qualitative research is necessary to gain a better understanding of the needs of the population and how factors not typically emphasised in smoking cessation interventions may be utilised. Understanding these perceptions is crucial to addressing the significant problem of tobacco use in this particular population which may require a more specialised approach to smoking cessation. This study seeks to give voice to the largely neglected population in drug treatment research by exploring service users' perceptions of smoking cessation in residential rehabilitation; to gain better understanding and identify ways to inform future smoking cessation intervention development in this setting.

## Methods

A generic qualitative approach [20–22] using semi structured, face-to-face interviews, useful for understanding service user perspectives, was adopted. This allowed participants to express their views in their own terms. Interview questions were informed by the literature and a pilot

interview was conducted to assess the suitability of the interview structure. A purposive sampling method [23] was chosen to reach participants who were, or had been, in residential treatment for substance misuse. Ethical approval for this study was obtained from a university research ethics committee.

All residents at two treatment services in the North West of England were invited to take part in the research, as were former clients who were still accessing the community treatment service.

The treatment services included were based on a therapeutic community model [24] and involved a voluntary 3–9 month admission to a group based program.

Eligible individuals were identified by staff and given participant information sheets prior to any engagement with the researcher. After one week the researcher returned to the services to answer any questions and to ascertain who was willing to be involved. Interviews were then scheduled at a mutually convenient time. Formal written consent was obtained prior to each interview. Information on local smoking cessation services and treatment providers, as well as different methods of cessation was made available at the end of the interviews.

Ten audio-recorded interviews lasting approximately 45 minutes each were carried out by the first author (ZS) at the treatment services, during staffed hours. Interviews were transcribed verbatim, removing any identifiable data, imported into NVivo 9 and analysed thematically [25]. Transcripts were open coded line by line. Codes were then grouped into categories and examined for salient themes. Saturation was considered to be reached as no new codes were identified in the final transcripts analysed. To establish trustworthiness of the analysis, a sample (n = 3) of the transcripts were checked by a member of the research team (LP). This involved the cross checking of coding strategies and the interpretation of data. The main themes identified are summarised below with verbatim quotes to illustrate findings.

## Results

### Profile

Of the ten participants (7 male; 3 female), four were currently in residential treatment, and one was in follow on treatment in the community. The other five had previously been in residential treatment (ranging from 1 to 10 years) and were currently volunteering or working in the community substance misuse service. Although the services included in this study were for the treatment of any substance use, of the participants in this study seven were in treatment for alcohol use only, the remaining three used both alcohol and drugs. Half (n = 5) were current smokers of tobacco, two were former smokers who were exclusively using e-cigarettes and three did not smoke at all. Three participants were aged between 30 and 40 years, four were between 40 and 50 years, two were between 50 and 60 years, and one was over 60.

### Themes

**An ideal opportunity for cessation.** There was a general consensus amongst participants that residential rehabilitation was a good time and place to begin the process of smoking cessation, as residents were already in the appropriate mind set for change, and that promoting a healthier lifestyle was important.

*Rehab is designed so that people can think about every aspect of their life, and smoking's one of them. You've got them there for six months, you might as well take advantage of the opportunity, it's a captive audience. . .Make them a gang of people that supports each other not to smoke. (male, smoker)*

*When you're in somewhere like that you can feel yourself changing, people are embracing change, you know. It'd be a good time to do that, get rid of both while you can (male, smoker)*

Five participants felt that the group setting was ideal and could benefit non-smokers too by focusing on wellbeing and diet as well as smoking. Three thought that smoking cessation was not the role of the treatment service and would be better offered as aftercare or as part of an ongoing treatment plan. There was also some concern that focusing on smoking cessation might jeopardise recovery from alcohol misuse.

*I think these things take time, and you should deal with one thing before the other. But after a length of time, if they were to put it to me, say "hey, well done, you've done well with drinking, how about having a look at smoking cessation? We'll be doing it here, you know the groups and what we've gone through", stuff like that, because I'm around people that I . . .trust, and trust is a massive thing, I would probably say yeah, why not. (male, smoker)*

**Facilitators for smoking cessation.** Several factors that promoted smoking cessation in the context of residential rehabilitation were identified. Four participants cited the financial cost of smoking as the primary reason they had or intended to quit or had switched to vaping or rolling tobacco:

*Obviously cost has quite a bit to do with it, and a lot of people around me when I was at (community drug and alcohol service) were on rollies, so I had one or two and I actually found that I preferred them. It's nice that the cost is phenomenally cheaper to buying packets of cigarettes. (male, smoker)*

Interestingly, the cost of cigarettes also contributed to some resentment. Services give residents a shared household budget and some participants observed that smokers would spend an unequal amount on tobacco, leaving less to buy the weekly food.

*They used to do community shops and two or three shops a week, so as a community we'd all get together. If you can imagine there's 20-odd people, some of that shop money was about buying cigarettes and papers. (male, former smoker)*

Despite universal awareness of the health consequences of smoking, five participants were current smokers and none mentioned health as a reason for quitting. Two had witnessed family or friends suffer from smoking related illnesses, and one former smoker had undergone surgery to remove a growth, which he thought was smoking related:

*I had a lump in my neck. . . the day I went into the hospital and the day before I went to hospital I smoked, and I came out of hospital and I've not had a cigarette since. (male, former smoker)*

**Challenges to smoking cessation in residential rehabilitation.** Whilst participants were generally positive about the prospect of smoking cessation in residential rehabilitation, some recognised that specific issues related to treatment services such as the shortage of staff time and resources and the attitudes of staff that smoking was of secondary importance would potentially act as barriers to cessation:

*It's such a secondary thing to what they're saying. . . it goes alcohol, then big list of drugs, with smoking at the bottom. So it's almost an afterthought really.* (male, smoker)

Other factors that could potentially hinder smoking cessation efforts in residential rehabilitation were highlighted. These included the perceived social and psychological benefits of smoking, the normalisation of smoking, policy and regulation, as well as perceptions and prioritisation of risk.

**Smoking as a social and psychological lubricant.**   Seven participants mentioned the social and psychological benefits of smoking, saying that it helped them cope with feeling socially anxious, especially when entering treatment. Some had started smoking as a way to fit in with others, to alleviate the boredom and the stresses of giving up alcohol.

*I didn't smoke at all, when I came here. When I was drinking heavily. So literally the day I got here I started smoking again. It was something to do. And I suppose, it's talking to people socially when you come in. And it's stopping drinking as well.* (male, smoker)

Five participants also cited that staff smoking with residents was a way to connect and engage with them; smoking acted as social lubricant.

*There's nowhere to hide when you're standing next to someone having a fag, it's more honest, more relaxed. It's the same as having a drink with someone really; it's a social lubricant, just in a different way. . .you're both indulging in the same semi-illicit activity, so it's easy to build up that bond of trust.* (male, smoker)

One participant reported that smoking had helped him get through difficult situations relating to his alcohol use and his poor mental health because it provided an escape.

*In times of stress when you're in there and you've got this feeling of, I need a drink, or you can visualise a bottle in front of you, you just think, I know, I'll go downstairs, I'll go outside and I'll have a fag, that will calm me down. And it does actually calm you down. That's how I feel, that's how a smoker feels, it calms you down. It takes you away from . . . what you're trying to deal with there* (male, smoker)

Another described his relationship with cigarettes and alcohol as a '*love affair'*, saying that despite the known harms of both he continued to use them because of an emotional attachment. Holding on to the secondary addiction (tobacco) made giving up alcohol feel like an achievable challenge. Tobacco was seen as the less immediate risk and less damaging, at least in the short term:

*It's an emotional wrench, taking alcohol away from someone, it certainly was for me. I was taking away something I loved, I mean my life was in a complete mess but I loved alcohol, but I knew it had to go. Taking away smoking at the same time. . .I would have thought twice about going into residential treatment.* (male, smoker)

**The culture of smoking in treatment services.**   Social norms and behaviours play a large part in smoking habits. When placed in a group setting, such as residential treatment, these norms become amplified, and the desire to fit in to a new setting can be perceived as essential to successful treatment outcomes. Participants widely regarded smoking in residential services as the norm.

*Over two thirds of any group. . .were smoking. . . and I'd say possibly half the staff smoked. It's kind of like a social gathering. There was no positive role modelling in regards to smoking, it was just the norm. (male, former smoker)*

Three non smokers felt excluded because they did not smoke, even if they chose to join smokers in the smoking area. One was frustrated at the way tobacco was normalised within the treatment service:

*I'm unusual because I haven't got a fag in my hand. I wish I did sometimes so that I could be part of the social group. (male, non-smoker)*

*Addiction's addiction, and if you minimise one against the other it kind of makes it alright, "you can't use substances here, you can't drink, but you can smoke all you want". (male, former smoker).*

Non-smokers who did not wish to sit in the smoking area felt socially penalised for not smoking.

*They were getting preferential treatment, the smokers, in that they (staff) were engaging with them more (female, non-smoker)*

The normalisation of smoking, as well as the fact that many staff were smokers, was seen to contribute to the continued and even increased tobacco use reported by some participants. Such entrenched normative behaviour would likely hinder any cessation interventions implemented.

**Relationship between tobacco and other substances.**   Smoking tobacco was viewed by participants as a very different issue to alcohol or illicit drug use, due to the perceived lack of intoxicating effect and its legal status. Three participants who claimed not to smoke still used tobacco in joints, illustrating that cannabis smoking and tobacco smoking were seen as different. Three also reported that tobacco use had increased with cannabis use. Many reported that they smoked more when drinking, one saying he could not have a drink without a cigarette. Even occasional or self-defined social smokers reported being more likely to smoke when having an alcoholic drink. All of these factors are reasons why people continue to smoke, and are resistant to stopping, and demonstrate the importance of addressing smoking in conjunction with use of other substances.

**Perceptions and prioritisation of risks.**   All participants acknowledged the health risks of smoking, but their perceptions and prioritisation of these risks varied. Notably, two of the younger participants did not feel that smoking was an immediate risk, believing that as long as they stopped at some point in the future, they could avoid damage to their health [26]:

*I know it's just another addiction, but for me, smoking wouldn't kill me as quickly as the amount I was drinking was going to. (male, smoker)*

The sense that other substances presented a more immediate risk and therefore should be prioritised was highlighted by eight of the participants:

*When you're dealing with people who are injecting heroin and crack into their groin four or five, six times a day, then you tend to gloss over the cigarettes, 'cause you know it's not going to kill them immediately, is it? (male, smoker)*

**Policy with treatment services.** Within the treatment services, policy changes had shifted smoking from inside, in communal living areas, to outside. Both services included in this study still allowed smoking in a designated area in the garden of the property. Whilst smoking policies were widely regarded as positive by non-smokers, there was concern that any further attempts to control tobacco use (such as banning it on the premises) would lead to conflict and tension. Even the non-smokers felt that enforcing a non-smoking policy was unfair and would be seen as a punishment.

> *I think people would go mad. Actually, literally. You know, they've got one little vice and maybe after they've cleared up the first one, once they've come out you can tackle it then, but I just don't think two in one go is very wise and for health reasons, you know, mentally and physically it's not good.* (female, non-smoker)

There were also concerns that any increased regulations would escalate behaviours such as illicit smoking in bedrooms or other residential spaces, which would increase the risk of fire. Additional worries that other substances might be smoked instead, and that smokers may become more irritable if denied access to cigarettes were also expressed:

> *I think they'd get a lot more people trying to smoke in their rooms, or you'd get a lot more illegal things being smoked because they can't go to the shop and buy cigarettes.* (female, non-smoker)

> *I have known groups that have scrapped the smoking break, and they don't tend to work as well. You tend to get a bit of animosity between people. The people who smoke, you can see them getting agitated when they're not quite sure when the break's coming.* (female, non-smoker)

In summary, the findings of this study included barriers and facilitators for implementing smoking cessation in this type of setting. Some of the challenges were posed by distorted perception of risk, staff attitudes and availability, established smoking culture in substance use treatment services, issues involving group living and the social aspects of smoking. Facilitators included the motivation to change and the social benefits of communal living, staff understanding and the rising costs of tobacco as a motivation to change smoking habits.

## Discussion

This study aimed to explore service users' perceptions of smoking cessation in residential substance use treatment. The study findings highlighted a general consensus that residential rehabilitation was an appropriate setting for smoking cessation although concerns that it might impede recovery were also cited, as suggested by existing literature [27]. A number of factors that would potentially help or hinder the delivery of smoking cessation in this setting were identified.

Supporting factors include acknowledgement of treatment as an ideal opportunity to make positive behavioural changes, with established group programmes and staff who understand the challenges of treating addiction. Substance use treatment and the associated motivation, social contact and staff with specialised knowledge and skills could all be utilised to capture the 'teachable moment' [10] and begin challenging individual and social norms associated with smoking. The reported increased use of e-cigarettes in the study population mirrors that of the wider population [28] and could be a facilitating factor in supporting their use as an approach

to smoking cessation, as recommended by NICE guidelines in the UK [29]. Although they remain controversial as a cessation tool [30], e-cigarettes allow smokers to retain their personal and social identity as a smoker, whilst reducing the harms associated with tobacco consumption. Whilst neither tobacco or e-cigarettes are safe and good for health, current available evidence suggests that vaping may be less harmful than tobacco smoking and therefore could be a better alternative.

Factors which act as barriers to smoking cessation in this setting were highlighted. Many participants reported that smoking helped them to deal with anxiety, stress and other highly emotional states, which supports previous evidence [31]. Entering residential treatment is associated with anxiety, as clients face the challenges of tackling their addiction but also the social conflicts associated with living with others. Social pressures added to the physical and psychological effects of substance withdrawal can heighten anxieties. Maslow's Hierarchy of Needs [32], often used in addiction counselling, can help to explain the distorted needs and priorities of someone with an addiction. In addiction, using and obtaining the substance becomes more and more of a priority, at the expense of anything else [33]. This can offer an explanation why awareness of the financial and health risks of smoking may be fully understood but still ignored. This theory can also be used to explore the perceived benefits of smoking, such as stress relief and social lubrication, which would need to be addressed in a smoking cessation intervention.

Understanding the clients' perceptions and their fears about giving up smoking could result in a better opportunity to engage with them and encourage positive change. It is also important to understand how giving up smoking could impact on an individual's anxiety and stress levels.

The emotional attachment of smokers to cigarettes has been mentioned, but it is also important to note that substance misuse treatment poses a challenge to the client's perceptions of self. The perceived risks of smoking were primarily cost rather than health related, which may be denial, the concept of reversibility of the damage done by continued smoking, or simply a reflection of immediate priority [34]. Consistent with the wider literature, particularly in younger participants, health risks were perceived as minimal, long-term smoking would be damaging but use for a short time was reversible. As other research suggests, other substances were thought to pose a greater risk than tobacco use, which may be a result of an increased knowledge of the damages done by drugs and alcohol and the focus on these substances in treatment services. Cheeta et al's work [35] reports a lack of perceived risks from tobacco in relation to other substances in a broader population.

Although participants mentioned health in some way, it was often a secondary factor. This is important to note when designing any future interventions, focusing on health risks alone will not be effective. The financial cost of smoking is known to be a significant factor is motivation to quit smoking, as well as to switch to e-cigarettes [36].

Despite recommendations that substance misuse and tobacco be treated together [7, 37], it was not evident at the two rehabilitation services included in this study. This may be due to organisational factors and cultural norms that do not recognize smoking cessation as part of the organisation's purpose [38], as well as budgets, limited resources and training for staff and the attitudes of staff [27].

Regardless of changes in tobacco policy over recent decades, many people in substance use treatment continue to smoke. It is therefore important to explore the reasons for this beyond regulation and policy. The limited evidence base on smoking cessation rates when done in conjunction with another substance presents a challenge, as long held beliefs in this field are that it is best to tackle one substance at a time [12]. This may act as/is a barrier to changing the smoking norms in this setting, as within the treatment setting, it is still commonplace to

smoke. Many participants reported this, and this has also been evidenced at a national level [39].

Residential treatment services could offer an ideal opportunity to integrate smoking cessation services and potentially improve outcomes for both substances, but further research and support from substance misuse treatment organisations is needed, as well as prioritisation from wider health organisations to ensure that smoking in specific sub populations is addressed. Public Health England have identified prisons, pregnancy, mental health and the NHS as priority populations [40], but have overlooked the substance using population.

## Limitations

This study included a small sample size, from a specific area within the UK, and as such the results may not be generalisable. Some participants had been in treatment several years previously, which may influence their perspectives. Further research and a larger sample size would be beneficial to address these issues.

## Recommendations

When considering future interventions, most participants agreed that attempts to enforce smoking cessation would be ineffective, whereas a more supportive and encouraging approach would be welcomed. This is reflected in existing literature which highlights the importance of psychological support as a factor in successful quit attempts [41].

Staff in residential and longer-term treatment services may be well placed to offer this support, and several participants mentioned that they would prefer to attend a service where they already felt comfortable with staff they know and who they feel understand their situation.

## Conclusion

The residential rehabilitation setting offers a unique opportunity to make positive lifestyle changes and reduce the health inequalities faced by people who use drugs and alcohol. Despite the many barriers discussed here, it is important that these are identified and considered when developing smoking cessation interventions to ensure effective and acceptable support. Whilst many of the perceptions expressed by participants are similar to those of the wider population, findings illustrate that this group has specific needs with regards to smoking cessation motivation and implementation, which need to be considered for any successful cessation interventions.

## Acknowledgments

The authors wish to thank the participants who gave their time and perspectives to inform this research, and without whom this would not have been possible.

## Author Contributions

**Conceptualization:** Zoe Swithenbank, Lorna Porcellato.

**Data curation:** Zoe Swithenbank.

**Formal analysis:** Zoe Swithenbank, Lorna Porcellato.

**Investigation:** Zoe Swithenbank, Lorna Porcellato.

**Methodology:** Zoe Swithenbank, Rebecca Harrison, Lorna Porcellato.

**Project administration:** Zoe Swithenbank.

**Supervision:** Rebecca Harrison, Lorna Porcellato.

**Writing – original draft:** Zoe Swithenbank.

**Writing – review & editing:** Zoe Swithenbank, Rebecca Harrison, Lorna Porcellato.

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
