## [Decision Letter · Decision Letter 0]

8 Feb 2022

PONE-D-21-31244Service user perceptions of smoking cessation in residential substance use treatmentPLOS ONE

Dear Dr. Swithenbank,

Thank you for submitting your manuscript to PLOS ONE. After careful consideration, we feel that it has merit but does not fully meet PLOS ONE’s publication criteria as it currently stands. Therefore, we invite you to submit a revised version of the manuscript that addresses the points raised during the review process.

We look forward to receiving your revised manuscript.

Kind regards,

Bronwyn Myers

Academic Editor

PLOS ONE

Journal Requirements:

2. Please note that in order to use the direct billing option the corresponding author must be affiliated with the chosen institute. Please either amend your manuscript to change the affiliation or corresponding author, or email us at plosone@plos.org with a request to remove this option.

Reviewers' comments:

Reviewer's Responses to Questions

**Comments to the Author**

1. Is the manuscript technically sound, and do the data support the conclusions?

Reviewer #1: Yes

Reviewer #2: Partly

2. Has the statistical analysis been performed appropriately and rigorously? 

Reviewer #1: Yes

Reviewer #2: N/A

3. Have the authors made all data underlying the findings in their manuscript fully available?

Reviewer #1: Yes

Reviewer #2: Yes

4. Is the manuscript presented in an intelligible fashion and written in standard English?

Reviewer #1: Yes

Reviewer #2: Yes

5. Review Comments to the Author

Reviewer #1: Thank you for asking me to review this manuscript. It is well thought out and well written. Please carefully consider by suggestions below. Please see attached document detailing some suggested comments and edits to strengthen an important study.

Reviewer #2: 1. Line 44: the word "centers" missing

2. There were a couple of references that were written in this pattern; Reid et al(Reid et al., 2008). This should be corrected throughout the manuscript

3. Lines 127-128: "Saturation was considered to be reached as no new codes were identified in the final transcripts analysed." I do not think saturation is achieved in analysis but interviews. I do not think this is the right use of the concept of 'saturation' in qualitative research.

4. Line 129: add the word "by" after "checked" in the sentence "were checked a member..."

5. Lines 134-140: please state why the participants were in rehabilitation? For Alcohol only, or other drugs or both?

6. Line 154: in a few places in the manuscript, no specific number of participants were mentioned. Since the number are so few, it would be good to see the number of participants replacing words such as "several" or "some".

7. Lines 143ff: Many of the headings do not seem to work well with the content in the results section. Consider reviewing the themes to more appropriately describe the content. For example "smoking is the norm in treatment centers" could be rephrased as "Smoking seen as a norm in treatment centers" Or "The culture of smoking in treatment centers"

8 Line 270: the methods section says there were only service users in the sample but this line contradicts that.

9. Please summarize the findings at the end of the results section

10. Lines 311 -316: the findings do not speak to harm reduction. Also, there is not enough evidence to make a claim that the user of e-cig in this sample did that to reduce harm. A change of addictive substance is not harm reduction.

11. Lines 312-313: sentence beginning in 312; I didn't see this in the results. Only saw a change of product to save money.

12. Lines 313-316: "Although they remain controversial as a cessation tool (Hartmann-Boyce et al., 2021), e-cigarettes allow smokers to retain their personal and social identity as a smoker, whilst reducing the harms associated with tobacco

consumption." The last phrase of this statement is not factual. There are several studies showing the harms caused by e-cigarettes. Though studies are on-going, there is presently sufficient evidence to confirm that e-cigarettes are harmful to health in many ways like traditional cigarettes.

13. Lines 340-343: I'm not sure this reason explains fully the issue presented here. I think awareness of the harms caused by tobacco and how soon it can affect them may be responsible. The greater risk of death in the long term may be responsible as opposed to visible immediate consequences of alcohol abuse. Cheeta et al 2018 as quoted in teh next sentence explains it better.

14. Lines 347-348: Please reference this statement or delete it. There is no evidence that rolling tobacco and illicit tobacco are more dangerous to health.

15. Add sections on limitations and conclusion

6. PLOS authors have the option to publish the peer review history of their article (what does this mean?). If published, this will include your full peer review and any attached files.

Reviewer #1: No

Reviewer #2: No

---

## [Author Response · Author response to Decision Letter 0]

24 Mar 2022

Please find that authors affiliations and the formatting of the manuscript have been amended as requested.

---

## [Decision Letter · Decision Letter 1]

3 Jun 2022

Service user perceptions of smoking cessation in residential substance use treatment services

PONE-D-21-31244R1

Dear Dr. Swithenbank,

We’re pleased to inform you that your manuscript has been judged scientifically suitable for publication and will be formally accepted for publication once it meets all outstanding technical requirements.

Kind regards,

Bronwyn Myers

Academic Editor

PLOS ONE

Additional Editor Comments (optional):

Reviewers' comments:

Reviewer's Responses to Questions

**Comments to the Author**

1. If the authors have adequately addressed your comments raised in a previous round of review and you feel that this manuscript is now acceptable for publication, you may indicate that here to bypass the “Comments to the Author” section, enter your conflict of interest statement in the “Confidential to Editor” section, and submit your "Accept" recommendation.

Reviewer #1: All comments have been addressed

Reviewer #3: All comments have been addressed

2. Is the manuscript technically sound, and do the data support the conclusions?

Reviewer #1: Yes

Reviewer #3: (No Response)

3. Has the statistical analysis been performed appropriately and rigorously? 

Reviewer #1: Yes

Reviewer #3: (No Response)

4. Have the authors made all data underlying the findings in their manuscript fully available?

Reviewer #1: Yes

Reviewer #3: (No Response)

5. Is the manuscript presented in an intelligible fashion and written in standard English?

Reviewer #1: Yes

Reviewer #3: (No Response)

6. Review Comments to the Author

Reviewer #1: Thank you for addressing my previous queries. The manuscript looks good and i feel that it is ready to publish. Congratulations to the authors.

Reviewer #3: (No Response)

7. PLOS authors have the option to publish the peer review history of their article (what does this mean?). If published, this will include your full peer review and any attached files.

Reviewer #1: **Yes: **Dr Lisa Dannatt

Reviewer #3: No

---

## [Editor Report · Acceptance letter]

10 Jun 2022

PONE-D-21-31244R1 

Service user perceptions of smoking cessation in residential substance use treatment 

Dear Dr. Swithenbank:

I'm pleased to inform you that your manuscript has been deemed suitable for publication in PLOS ONE. Congratulations! Your manuscript is now with our production department. 

Kind regards, 

on behalf of

Dr. Bronwyn Myers 

Academic Editor

PLOS ONE